# Fabrication of Active Polymer Optical Fibers by Solution Doping and Their Characterization

**DOI:** 10.3390/polym11010052

**Published:** 2018-12-31

**Authors:** Igor Ayesta, Mikel Azkune, Eneko Arrospide, Jon Arrue, María Asunción Illarramendi, Gaizka Durana, Joseba Zubia

**Affiliations:** 1Department of Applied Mathematics, Engineering School of Bilbao, University of the Basque Country (UPV/EHU), Plaza Ingeniero Torres Quevedo, 1, E-48013 Bilbao, Spain; eneko.arrospide@ehu.eus; 2Department of Communications Engineering, Engineering School of Bilbao, University of the Basque Country (UPV/EHU), Plaza Ingeniero Torres Quevedo, 1, E-48013 Bilbao, Spain; mikel.azkune@ehu.eus (M.A.); jon.arrue@ehu.eus (J.A.); gaizka.durana@ehu.eus (G.D.); joseba.zubia@ehu.eus (J.Z.); 3Department of Applied Physics I, Engineering School of Bilbao, University of the Basque Country (UPV/EHU), Plaza Ingeniero Torres Quevedo, 1, E-48013 Bilbao, Spain; ma.illarramendi@ehu.eus

**Keywords:** polymer optical fibers, rhodamine B, solution-doping technique, photostability, optical characterization, penetration of the dopant solution

## Abstract

This paper employs the solution-doping technique for the fabrication of active polymer optical fibers (POFs), in which the dopant molecules are directly incorporated into the core of non-doped uncladded fibers. Firstly, we characterize the insertion of a solution of rhodamine B and methanol into the core of the fiber samples at different temperatures, and we show that better optical characteristics, especially in the attenuation coefficient, are achieved at lower temperatures. Moreover, we also analyze the dependence of the emission features of doped fibers on both the propagation distance and the excitation time. Some of these features and the corresponding ones reported in the literature for typical active POFs doped with the same dopant are quantitatively similar among them. This applies to the spectral location of the absorption and the emission bands, the spectral displacement with propagation distance, and the linear attenuation coefficient. The samples prepared in the way described in this work present higher photostability than typical samples reported in the literature, which are prepared in different ways.

## 1. Introduction

In the last few years, interest in the field of photonics has increased as a result of the incorporation of functional materials into solid-state organic hosts, especially into polymer optical fibers (POFs) [1,2]. As compared to glass fibers, POFs are easier and more economical to manufacture, safer to handle, and much more flexible [3,4]. As the manufacturing temperature of POFs is much lower than that of glass fibers, it is possible to embed a wide range of dopants in the fiber core, from organic dyes and conjugated polymers to other kinds of materials, such as rare-earth ions and quantum dots. Some of these dopants can generate or amplify visible light at the low-attenuation windows of POFs [5,6]. The emission and absorption features of these dopants can be suitable for achieving luminescence in the visible region of the spectrum, which is interesting for a wide range of applications [7,8,9]. Consequently, research on doped POFs is on an upward trend. The results obtained are being applied for the development of optical sensors [7,8], of solar concentrators that collect and transport solar light [9], and of superluminiscent speckle-free light sources [5,10]. Additionally, the use of optical fibers allows for symmetrical output beams, due to their circular symmetry, and for longer interaction distances between light and dopant, owing to the guidance of light. Moreover, the confinement of light as it propagates facilitates connectivity, and the high ratio between surface area and volume in the fiber improves the heat-dissipation efficiency.

There are different approaches to incorporate the dopant molecules into the Poly(methyl methacrylate) (PMMA) core. Commonly, the doping is carried out at the pre-polymerization stage. The dopant molecules are added to the monomer mixture before carrying out the polymerization process [11,12]. Alternatively, the dopant molecules, together with the melted PMMA, can be mixed during the extrusion process carried out to fabricate the POF preform [13]. The third approach consists in introducing the dopant molecules into the microstructured fiber preform by using the solution-doping technique [14]. However, all these approaches are performed at the fiber preform level, which means that a fiber-drawing tower is needed to obtain the doped POFs. These towers are very expensive, so not all researchers can afford to have one. In contrast, this work describes a doping procedure, based on the solution-doping technique, in which the incorporation of dopant molecules into the core material is carried out directly into the fiber, instead of into the preform. This procedure also opens up the opportunity for any research group to prepare their own ad-hoc doped POFs for any specific application requiring precise characteristics such as distinct dopant molecules or fiber diameters. To date, few works employing the solution-doping technique in POFs have been reported, and only the preform or the fiber cladding were doped [14,15]. In [15], cladded POFs were employed. The dopant molecules could not thereby penetrate into the core, because the interface between the core and the cladding acted as a barrier for them, while the solvent (methanol) could penetrate. Therefore, the cladding was the only doped region. However, following the procedure described in this paper, we can obtain POFs whose core is completely and uniformly doped with active molecules. The dopant utilized for this work is the organic dye rhodamine B. As far as we know, this is the first time that this procedure and its optical characterization have been reported.

The paper is organized as follows. First, the method for preparing the samples is explained. Then, the experimental setup is described, together with the penetration characteristics of the solution of methanol and dopant into the fiber samples. Afterwards, the optical characterization is shown and discussed.

## 2. Materials and Methods

### 2.1. Sample Preparation 

The POF samples employed in this work were fabricated by our research group from Plexiglass^®^ extrusion rods purchased from Evonik (Essen, Germany). These were 20 mm in diameter and had a weight-average molecular weight (*M*_w_) of 110301. They were annealed in a C-70/200 climate chamber (Controltecnia-CTS, Hechingen, Germany) for 7 days at a temperature of 90 °C under low relative humidity. Afterwards, they were drawn into only core fibers of 500 μm in diameter using our own POF drawing tower, which enabled us to control the fiber diameter with an accuracy of ±1% [16]. The refractive-index profile of the fibers was step-index in all cases. Alternatively, there are only core POFs commercially available, which could also be employed for the preparation of doped fibers.

To incorporate rhodamine B dopant molecules into the pristine only core fibers, we cut samples of around 5 cm and put them in an oven at a temperature of 60 °C in order to eliminate any possible residual stress [17]. The solution employed in this work was prepared by dissolving 500 mg of rhodamine B (Merck kGaA, Darmstadt, Germany) in 200 ml of methanol (Thermo Fisher Scientific Inc., Madrid, Spain). As will be shown in this paper, the resultant dopant concentration was high enough for us to visualize the penetration of the solution into the samples. Rhodamine B is a well-known and widely used dopant for optical-communication purposes because its emission spectrum is located in one of the low-attenuation windows of the PMMA material [18]. In any case, other dopants of similar size, such as different rhodamines or fluorescein, could also be employed in our solution-doping technique. The reason for utilizing methanol as solvent is that rhodamine B can be easily dissolved in it, but not the PMMA. This property prevents the formation of dopant aggregates, which has detrimental effects on the light-emission features of doped fibers [19]. In addition, the maximum immersion time of the host PMMA material in order for it not to be damaged by the solvent is larger with methanol than with other solvents [17]. Moreover, methanol is also quite volatile, which allows it to be removed from the PMMA below its glass transition temperature [14]. The solution prepared in this way was employed for different samples by filling multiple small laboratory glass bottles with the solution and introducing samples that were to be treated at the same doping temperature in each bottle. Afterwards, these bottles were kept at different temperatures, depending on the desired doping temperature: in a fridge (10 °C), in a chamber at room temperature (20 °C), or in a laboratory water bath (from 35 °C to 50 °C). After the desired time of thermal treatment for each bottle, the samples were taken out of the solution, thoroughly rinsed and dried at room temperature. Then, a close look at the cross-sections of these samples through an optical microscope allowed us to monitor the different lateral penetrations of the solution of solvent and dye. To acquire images of these cross-sections, each sample was cut at a distance of around 3 mm from one of its ends, which was polished by hand using polishing paper. Each sample was then connectorized with fiber-chuck connectors and placed on the sample holder of the optical microscope in an upright position, in order for the image of its cross-section to be easily taken.

Some of the doped samples obtained in this way were then cladded with a layer of around 10 microns. For this purpose, the only core doped samples were immersed once for 30 s at room temperature in a solution containing the material of the cladding. This consisted of the polymer PC-404PF (Luvantix ADM, Daejeon, Korea) in isopropyl-alcohol at the concentration of 1/3 *v*/*v*. Finally, the cladded samples were cured with UV radiation.

### 2.2. Experimental Setup

Figure 1 sketches the experimental setup employed to measure the optical spectra by means of the side-illumination technique [20,21]. The POF sample is excited laterally and the light emitted from the excitation point propagates along the sample over a distance that can be adjusted by moving the linear stage employed to hold the sample. The light is finally collected by an optical spectrometer placed at the end of the sample. This is the procedure followed for all the measurements carried out as functions of either the propagation distance or the excitation time. The excitation light source was a Mai Tai HP laser system (Spectra-Physics, Newport, Santa Clara, CA, USA), which emitted Gaussian light pulses of around 100 fs at a repetition rate of 80 MHz, with peak powers of 300 kW and an average power of 2.5 W. The spot of the laser beam was 1.2 mm in diameter. The wavelength of the original light was reduced by using a second harmonic generator (Inspire Blue, Radiantis, Barcelona, Spain) in order to be able to excite from 345 nm to 520 nm. The excitation irradiance was controlled by placing a variable attenuator after the laser output and adjusting it by hand.

Each of the doped POFs to be measured was held in place with xy-micropositioners standing on a linear stage, thus allowing for the fiber to be maintained completely horizontal while centering the incident laser beam on the fiber symmetry axis. For all of the fiber samples, the spectrum emitted from the polished fiber end was measured by means of a USB4000-UV-Vis spectrometer (Ocean Optics, Largo, FL, USA) with an optical resolution of 1.5 nm for the full width at half-maximum (FWHM). All the obtained data were corrected for the response of the detection system. Additionally, a reference signal was taken using a beam splitter in order to cancel out any power fluctuation of the laser. By changing the position of the laser-excitation point, the variation of the emission spectrum as a function of the propagation distance through the doped fiber was also analyzed. For this purpose, an ILS250CC (Newport, Santa Clara, CA, USA) linear stage driven by an ESP300 (Newport, Santa Clara, CA, USA) motion controller was utilized. The whole acquisition system was automated by means of a LabVIEW program elaborated by us.

Additionally, the absorption spectra of the doped fibers were measured by a Cary 50 UV-Vis spectrophotometer (Agilent Technologies, Santa Clara, CA, USA). In these measurements, the length of the fibers was short enough (1 cm) for the absorption bands of the dopant to be detected.

## 3. Results and Discussion

### 3.1. Penetration of the Solution

Figure 2 shows cross-sectional images of fiber samples that were immersed in a solution of rhodamine B and methanol at 10 °C for different time intervals. The lateral penetration of the solution is clearly visible, because of the clear boundary between the outer ring, which is swollen and orange, and the intact transparent inner region. As the immersion time increased, the solution penetrated deeper into the fiber until full penetration was reached. Although Figure 2 only shows the progress in the penetration process corresponding to 10 °C, the process of formation of an outer ring is analogous at other temperatures, but the dynamics are different in relation to its evolution with time. 

To assess the penetration of the solution into the fibers, the ring’s front thickness and the relative dopant penetration were considered. The latter is defined as the quotient (*φ_out_* − *φ_in_*)/*φ_out_*, where *φ_out_* is the outer diameter of the fiber and *φ_in_* is the diameter of the inner unpenetrated region [13]. Both parameters were calculated for three samples doped under the same conditions, and each experimental point of Figure 3 shows the corresponding average value.

According to the relative values of penetrant mobility and segment relaxation of the polymer, the diffusion behavior can be categorized into three different cases [21,22]. These are denoted as follows: Case I, or Fickian diffusion, in which the penetrant mobility is much lower than the segment relaxation of the polymer; Anomalous Case, in which the penetrant mobility is similar to the segment relaxation; and Case II, or non-Fickian diffusion, in which the penetrant mobility is much larger than the segment relaxation. The latter case features a sharp boundary between the outer swollen layer and the inner core material, which advances with uniform velocity. In all these cases, the front penetration thickness (*d*) is related to the immersion time (*t*) through the following expression [23]:(1)$$d=k\cdot t^{n},$$
where *k* is a constant and *n* is a real number that varies from case to case: usually 0.5 for Case I, 1 for Case II, and any intermediate value for the Anomalous Case.

The results presented in Figure 3 and in Table 1 demonstrate that the penetration behavior of our solution of rhodamine B and methanol into our POF samples gradually changed from Case II to Case I as the temperature increased. For low temperatures (10 °C and 20 °C), both the relative dopant penetration and the front thickness advanced at constant speed. Moreover, the exponent n was equal to 1, which means that the category of the diffusion behavior was clearly Case II. For higher temperatures (above 20 °C), the advance of the front thickness was much faster than before. However, the diffusion of the solution through the swollen layer could no longer keep a constant concentration at the front, thus producing a slight deceleration in the advance of the front as the penetration approached the center of the fiber considered. Consequently, the value of *n* shifted towards 0.5, which is characteristic of Case I.

At first glance, the use of high temperatures seems to be more interesting from the point of view of the time required to complete the doping procedure. However, as will be shown in the next section, the optical characteristics would be negatively affected at doping temperatures that are close to or above room temperature.

Figure 4 shows the Neperian logarithm of the penetration rates obtained from Figure 3 plotted against the inverse of the absolute temperature (i.e., the Arrhenius plot), where T is measured in kelvins. The slope of the fitting of the points to a straight line is proportional to the activation energy for the occurrence of the diffusion. In our case, this energy is 27.64 kcal/mol, which is close to the values of some PMMA films immersed in methanol (27 kcal/mol was reported in [24]). This means that the activation energy is not significantly affected by the presence or not of the dopant or by the geometry of the PMMA system.

### 3.2. Optical Characterization

The analysis of the doping diffusion at different temperatures revealed that the translucency of the doped material was much lower when the doping process was carried out above room temperature (i.e., at any of the temperatures equal to or greater than 35 °C in Figure 3), foreshadowing bad light-transmission features. This effect is clearly noticeable in Figure 5, and it is a direct consequence of the interaction between the methanol solvent and the PMMA of the fiber core. It can be explained as follows: the glass transition temperature (*T*_g_) of a polymeric material can change when it is immersed in a solvent. In fact, the *T*_g_ value of the polymer can be reduced down to room temperature if it is immersed in methanol [25]. Specifically, the *T*_g_ for a methanol-PMMA system can range between 20 °C and 30 °C, depending on the weight-average molecular weight (*M*_w_) of PMMA (*T*_g_ = 20 °C for *M*_w_ = 23,500 g/mol, and *T*_g_ = 30 °C for *M*_w_ = 550,000 g/mol) [26]. As we mentioned in the section about the sample preparation, the value of *M*_w_ for our fibers is 110301, which means that the *T*_g_ of our samples immersed in methanol should be only slightly higher than 20 °C. When the temperature of the doping process is higher than the *T*_g_, as was the case in some of our experiments, small holes are formed in the PMMA material. Moreover, as the doping temperature is increased, the size and the population of these holes increases, which raises the optical attenuation of the fibers dramatically [27]. This effect can be clearly seen in Figure 6, in which the attenuation coefficients were measured by the spectrophotometer for different fiber samples immersed in methanol at the temperatures mentioned in Figure 3. A complete methanol penetration was achieved in all cases. For fibers doped at 10 °C, the optical attenuation was not much affected by the immersion, obtaining almost the same attenuation values as those of pristine fibers. This suggests that holes could not grow because the immersion temperature was still below the corresponding *T*_g_ value. However, at 20 °C, the attenuation began to be affected, especially in the visible part of the spectrum. As for the rest of the temperatures, i.e., above 35 °C, measurements could not be carried out, because the light attenuation was too high.

Heating the immersed fiber above its *T*_g_ value also has consequences on the relaxation of internal stresses in the PMMA material. As a matter of fact, the relaxation is usually followed by changes in the physical dimensions, such as shrinkages in the axial direction as well as growths in the diameter [28,29]. We observed these effects even at 20 °C, so we can conclude that the most appropriate doping temperature is around 10 °C for the fibers employed in this work.

Taking the aforementioned results into account, samples of 12 cm were fabricated at the suitable temperature of 10 °C. Moreover, in a second stage we incorporated a cladding layer of around 10 µm of a polymeric material to these optimum samples, in order to improve the light transmission features. Additionally, the core material is less prone to being affected by external unwanted agents if a cladding is added. This process was carried out in the way described in the section about the preparation of the samples. The absorption coefficient and the emission spectrum of these samples in the near-ultraviolet and visible regions are shown in Figure 7. The presence of the dopant is apparent in the prominent absorption band peaking at 558 nm. The emission spectrum was measured by exciting the sample at 520 nm with 22 nJ/cm^2^ of irradiance. Its maximum value is located at 592 nm. Actually, the peak wavelength is affected by the red shift produced by reabsorption and reemission processes along the connectorized initial length of the fiber [30], so shorter emission wavelengths would be expected if the spectrum were measured in a thin piece of bulk material. In any case, the obtained absorption and emission peaks are very similar to those reported for PMMA fibers that were also doped with rhodamine B, but utilizing different doping techniques, such as the interfacial gel polymerization [31,32]. We have estimated that the dopant concentration of our doped fibers is in the range between 1 and 10 ppm, by using the method reported in [33], which is based on the red shift of the emission as the fiber concentration is varied.

The evolution of the emission spectra obtained for each propagation distance z along the fiber is illustrated in Figure 8, which corresponds to our POF doped with rhodamine B at 10 °C. These spectra were not symmetrical, so their evolution is characterized by means of their first and second moments, *N*_1_ and *N*_2_, respectively. The first moment represents the average emission wavelength, and the square root of the second moment is proportional to the spectral width. As is well known, separating the excitation point on the fiber away from the detector causes a reduction in the emitted irradiance, while the spectrum is shifted toward longer wavelengths, with no significant changes in the spectral widths [19]. The spectral behavior as a function of the propagation distance shown in Figure 8 for our fibers is the same as that reported for other doped fibers manufactured using other techniques such as the interfacial gel polymerization. Please note that these kinds of spectral shifts could be employed for applications such as the manufacture of tunable light sources or the design of displacement sensors based on doped fibers [34].

The attenuation of each of our doped fibers was calculated by measuring the decrease in the irradiance of the fluorescence spectra as the light propagation distance z increased. Assuming that the illuminated fiber section behaves as a plane-wave source, the light irradiance propagating towards the photodetector at any wavelength *λ* decays exponentially with z as follows [20,21]:(2)$$I(\lambda ,z)=I_{0}(\lambda )\cdot \exp(-\alpha (\lambda )\cdot z),$$
where *I*_0_(*λ*) represents the light irradiance measured at *z* = 0 at the wavelength *λ*, and *α*(*λ*) is the linear attenuation coefficient at the considered wavelength. Figure 9 shows the linear-attenuation coefficients calculated for our doped fibers by fitting the experimental data to Equation (2) for several wavelengths [35]. The attenuation values obtained in the way described above are very similar to those reported in the literature for other doped PMMA POFs, including thermoplastic ones and even thermosetting ones doped with rhodamine 6G [36]. This implies that the doping procedure employed in this work did not add extra losses to the fibers with respect to the typical attenuation values of fibers fabricated by means of more conventional techniques, such as the interfacial gel polymerization. This holds true as long as the doping temperature is maintained below the corresponding *T*_g_. This result, together with the previous ones, serves to validate the doping technique employed in this work.

Finally, the photostability of the emitted spectra was also studied for our doped samples by pumping them at 520 nm with our femtosecond laser at an irradiance of 22 nJ/cm^2^. The fibers utilized for this purpose had not been previously exposed to laser light, in order to avoid previous degradation that would have affected the results. The relative fluorescence intensities were normalized to 100% at the start of the measurements. As shown in Figure 10, the fluorescence capacity of the doped fibers was reduced by 32% after 60 min of exposure, which, in this case, was equivalent to 2.88 × 10^11^ laser shots. This reduction is smaller than those reported for other POF samples doped by means of the polymerization technique when subjected to the same exposure conditions [19,36]. Specifically, the photostability of our doped fibers was, respectively, 7% and 18% higher than those measured in the same conditions for thermoplastic and thermosetting POFs doped with rhodamine 6G. Moreover, as Table 2 shows, our doped fibers also presented a greater photostability than that reported for POFs doped with some conjugated polymers, such as PFO, F8T2 or PF3T [19]. The improvement in the photostability can be explained as follows. While the gel-polymerization technique tends to form dopant aggregates in the fiber core during the fabrication process [37], the doping procedure employed in this paper is less prone to the formation of such aggregates. The low segmental-relaxation rates of the PMMA, at least when the temperature is low enough, does not allow the settling of many dopant molecules together in the fiber core, so individual dopant molecules tend to be more spaced. The consequent absence of aggregates reduces the number of non-radiative relaxations of the dye molecules and, therefore, prevents them from optical bleaching [38]. Moreover, as the dopant molecules are mainly surrounded by the PMMA material, these are partially protected from the thermally induced degradation effects, so the fibers tend to conserve their emission capacity for a longer time [39]. The fact that the improvement in photostability is larger in the case of the comparison with the result reported for the thermosetting fiber could also be related to the much greater dopant concentration employed in their case. Improvements in photostability were also observed in the case of rhodamine-6G PMMA films pumped with visible or ultraviolet radiation when they were doped by means of a diffusion procedure instead of the typical gel-polymerization technique [40]. However, this is the first time that this effect is observed in POFs, as far as we know, which could be very useful in the design and development of all-optical devices based on this kind of fibers with longer lifespans.

## 4. Conclusions

This paper shows a method of fabrication of active POFs starting from only core PMMA fibers, by immersing them in a solution of methanol and the organic dye rhodamine B for an appropriate period of time at a suitable temperature. We demonstrate that the penetration behavior gradually changes from the ideal diffusion category known as Case II to the one known as Case I or Fickian diffusion as the temperature is increased from low values (10 °C) to high ones (50 °C). Although the front penetrates more slowly at lower temperatures, the corresponding linear-attenuation coefficients are lower, due to the absence of hole formation in the core material if, and only if, the temperature of the doping process is lower than the corresponding glass-transition temperature. Moreover, we also measure and analyze the spectral characteristics and the intensity of the emission of doped fibers as functions of both the propagation distance and the excitation time. The corresponding results are in good agreement with those reported for similar active POFs doped by means of other fabrication techniques, such as the gel-polymerization technique. Furthermore, our fibers present a higher photostability, owing to the lower probability of formation of dopant aggregates when using our doping technique. This improvement, together with the rest of optical features and ease of fabrication, could be very interesting in the design process of all-optical devices based on active POFs, such as switches, lasers and amplifiers, optical sensors and solar concentrators.

## Figures and Tables

**Figure 1 polymers-11-00052-f001:**
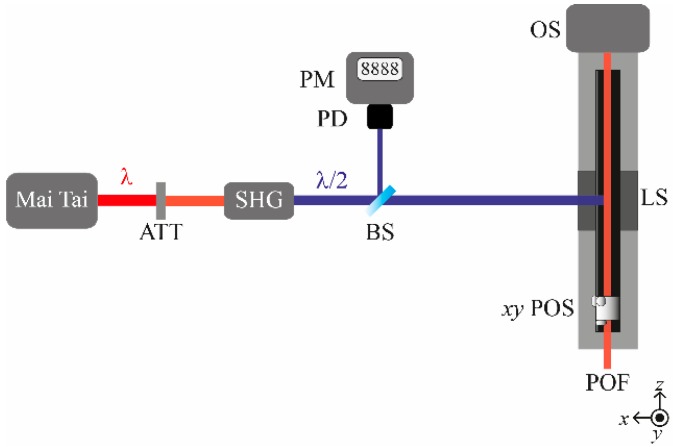
Experimental setup employed to measure the emission spectra of our doped POFs. Legend: ATT: variable attenuator; SHG: Inspire Blue second harmonic generator; BS: beam splitter; PD: photodetector; PM: power meter; LS: linear stage; *xy*-POS: xy-micropositioner; OS: optical spectrometer.

**Figure 2 polymers-11-00052-f002:**
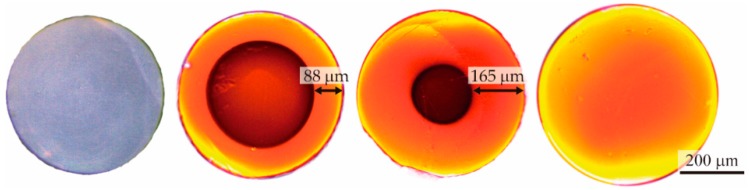
Microscope images of the cross-sections of POF samples immersed in a solution of rhodamine B and methanol at 10 °C for different time intervals, namely (from left to right): 0 h, 30 h, 54 h and 96 h. The image of the pristine fiber was taken in transmitting mode. The images of the doped fibers were obtained under lateral UV excitation.

**Figure 3 polymers-11-00052-f003:**
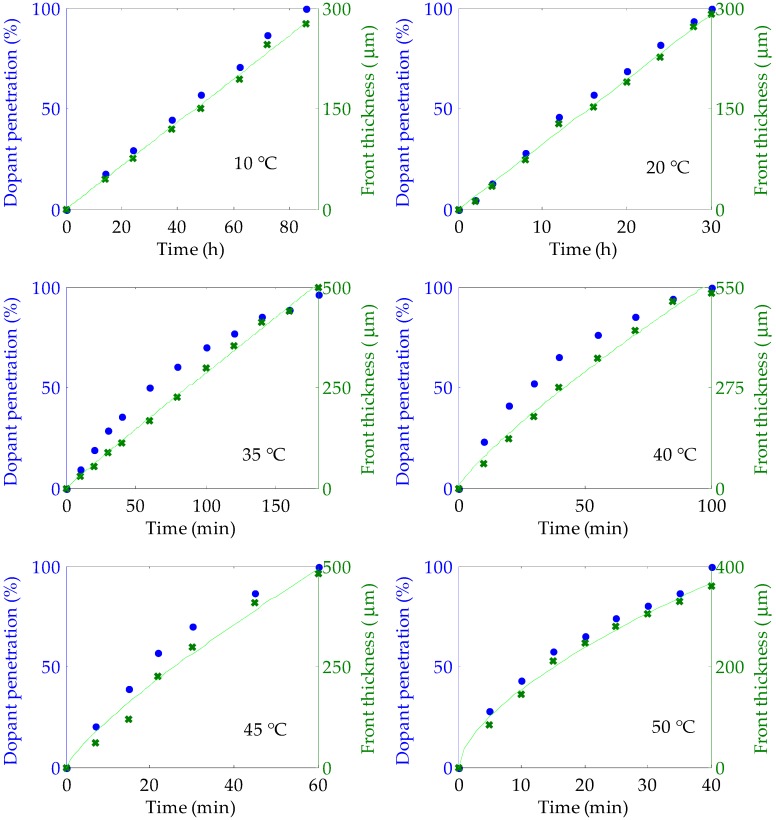
Temporal evolutions of the relative dopant penetration (blue circles) and of the ring’s front thickness (green crosses) for six different doping temperatures: 10 °C, 20 °C, 35 °C, 40 °C, 45 °C and 50 °C. The green lines are the fittings of the front thicknesses at each temperature to Equation (1).

**Figure 4 polymers-11-00052-f004:**
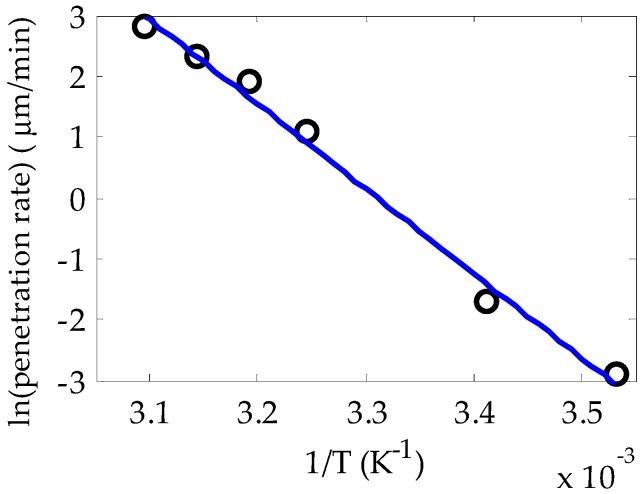
Arrhenius plot for the solution of rhodamine B and methanol penetrating in PMMA POFs. The coefficient of determination of the fitting is 0.9901.

**Figure 5 polymers-11-00052-f005:**
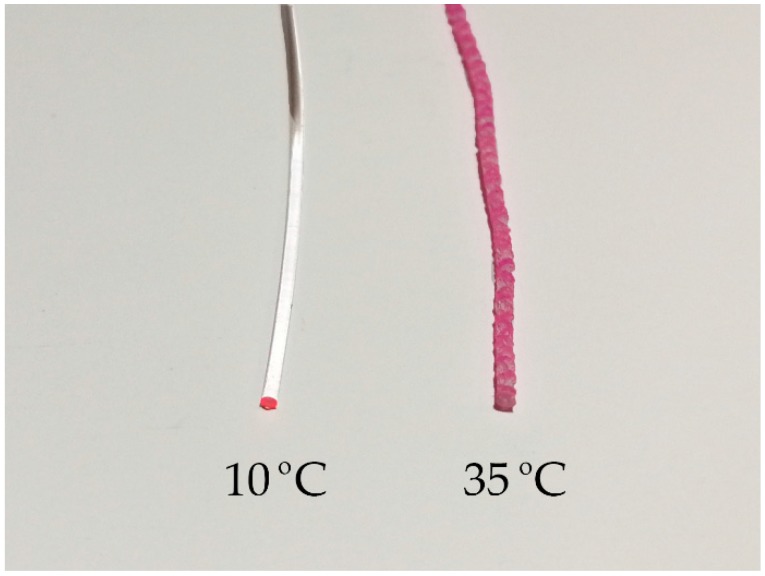
External appearance of two doped samples prepared at 10 °C and at 35 °C.

**Figure 6 polymers-11-00052-f006:**
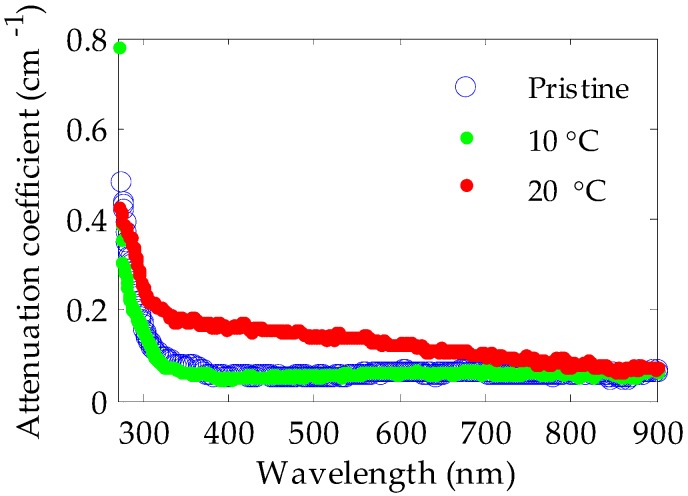
Attenuation coefficients measured for pristine PMMA POFs and for samples immersed in methanol at two different temperatures. At the temperature of 35 °C or above it the attenuation was too high for the measurements.

**Figure 7 polymers-11-00052-f007:**
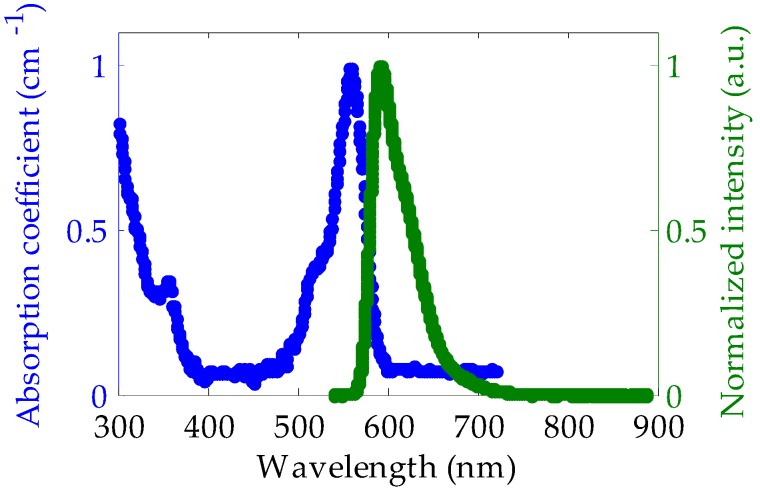
Absorption coefficient and emission spectrum corresponding to our POF doped with rhodamine B by using the solution-doping technique at 10 °C. The emission spectrum corresponds to an excitation wavelength of 520 nm and to a propagation distance of 4 cm.

**Figure 8 polymers-11-00052-f008:**
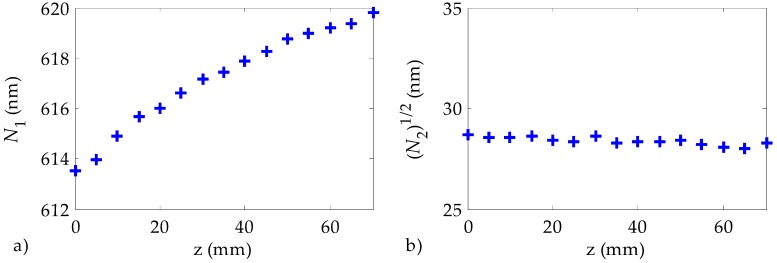
Evolution of (**a**) the first moment (*N*_1_) and of (**b**) the square root of the second moment (√*N*_2_) as a function of the excitation wavelength for our POFs doped with rhodamine B at 10 °C. The fiber was pumped at 520 nm with an irradiance of 22 nJ/cm^2^. The z of the point of the fiber closest to the detector is normalized to 0.

**Figure 9 polymers-11-00052-f009:**
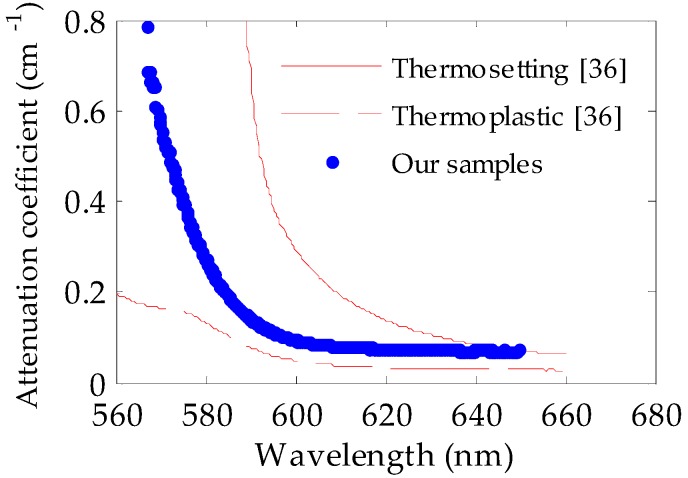
Linear attenuation coefficients of our POFs doped with rhodamine B at 10 °C (thick blue dots), together with those reported in [36] for thermosetting (thin solid line) and thermoplastic (thin dashed line) PMMA POFs doped with rhodamine 6G. The excitation wavelength and irradiance were 520 nm and 22 nJ/cm^2^, respectively.

**Figure 10 polymers-11-00052-f010:**
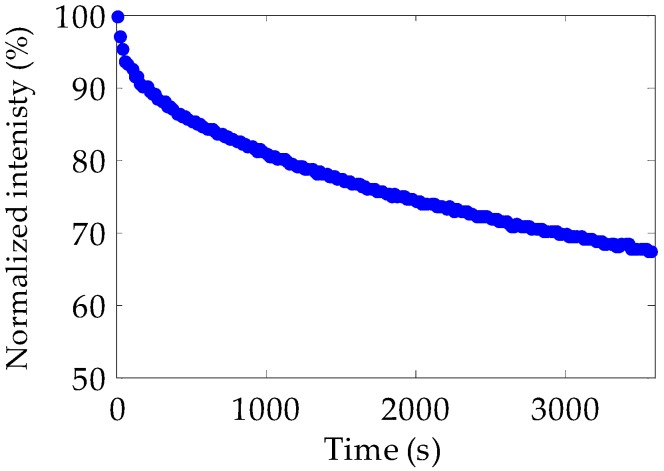
Fluorescence intensity as a function of the excitation time in our POF samples doped at 10 °C. The excitation wavelength was 520 nm, the excitation irradiance was 22 nJ/cm^2^, and the light propagation distance was 4 cm.

**Table 1 polymers-11-00052-t001:** Values of the fittings of the experimental front thicknesses of Figure 3 to Equation (1), and the corresponding coefficients of determination.

Temperature (°C)	*k*	*n*	*R*^2^
10	3.2 ± 0.1	1 ± 0.05	0.9961
20	9.6 ± 0.2	1 ± 0.05	0.9972
35	3.3 ± 0.9	0.97 ± 0.06	0.9972
40	12 ± 5	0.83 ± 0.09	0.9922
45	18 ± 12	0.81 ± 0.09	0.9835
50	37 ± 9	0.62 ± 0.07	0.9935

**Table 2 polymers-11-00052-t002:** Percentage rate of degradation of the fluorescence intensity for several doped fibers under the same excitation conditions: 60 min of excitation with 22 nJ/cm^2^ of irradiance.

Fiber Sample	Degradation (%)	Reference
Our samples	32	-
Thermoplastic fiber doped with rhodamine 6G	39	[36]
Thermosetting fiber doped with rhodamine 6G	50	[36]
PMMA POF doped with PFO	81	[19]
PMMA POF doped with F8T2	81	[19]
PMMA POF doped with PF3T	45	[19]

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
