# Peer review of "Fabrication of Active Polymer Optical Fibers by Solution Doping and Their Characterization"

_polymers, 2018, doi:10.3390/polym11010052_

Round 1
Reviewer 1 Report
The paper by Igor Ayesta presents fabrication of active polymer optical fibers by solution doping. However, there remains many points required to be more clear. So it can only be considered for publication provided the mandatory revisions that address my concerns below:
-At the abstract, the author uses ‘These features are in good agreement with those observed in the literature for similar active POFs.’. If possible, please be more clear about it.
-From the experimental results (Fig. 2), the approach adopted seems time consuming. Especially, it is a bit hard imagine that the influence of the solvent can almost be removed. So what is the advance compared with other method, like doping with extrusion diffusion or doping before polymerization? If possible, please be more clear about the motivation.
- If possible, please indicate the front thickness in Fig. 2.
-For the results of Fig. 3, it is better to use the similar scale for good comparison. In addition, please add the error bar for all the fitting results (similar for Fig. 4).
-Could you please give more information about how you carry out the experiment of absorption measurement? What is the length of the sample used for this experiment? BTW, where you cited Figure 5?
- It is not good to compare the efficiency by the wavelength. In fact, lifetime could be more related with the efficiency.
-in Line 293-295, the author claimed that the reduction is smaller than other POF samples. If possible, please listed as Table for better comparison.
-The English is casual at some places (such as ‘Poly(methyl methacrylate)’, ‘reference signal power was’, ‘it might seem that the’, ‘The so obtained’, etc). Some font size is not right. The author should revise it carefully and thoroughly.
-Some reference format/style is inconsistent (like 74, 9, 10, 23, 28). This should be revised and unified.
Author Response
Manuscript ID: polymers-405534
Title: Fabrication of active polymer optical fibers by solution doping and their characterization
The changes in the paper are indicated below in red. The comments to the reviewers are written just above the changes and after each recommendation.
The paper by Igor Ayesta presents fabrication of active polymer optical fibers by solution doping. However, there remains many points required to be more clear. So it can only be considered for publication provided the mandatory revisions that address my concerns below:
Changes following the recommendations by the reviewer:
1. At the abstract, the author uses ‘These features are in good agreement with those observed in the literature for similar active POFs.’. If possible, please be more clear about it.
Following the reviewer’s recommendation, we have added the following sentence in the abstract:
Moreover, we also analyze the dependence of the emission features of doped fibers on both the propagation distance and the excitation time. Some of these features and the corresponding ones reported in the literature for typical active POFs doped with similar dopants are quantitatively similar among them. This applies to the spectral location of the absorption and the emission bands, the spectral displacement with propagation distance, and the linear attenuation coefficient. The samples prepared in the way described…
2. From the experimental results (Fig. 2), the approach adopted seems time consuming. Especially, it is a bit hard imagine that the influence of the solvent can almost be removed. So what is the advance compared with other method, like doping with extrusion diffusion or doping before polymerization? If possible, please be more clear about the motivation.
We agree with the reviewer that the method employed in this paper to obtain doped POFs could be more time consuming than some typical techniques, such as the polymerization gel technique or the extrusion one. However, the main disadvantage of these typical techniques is that, in all cases, they require a fiber-drawing tower, because the doping process is carried out at the preform level. These towers are very expensive, so not all researchers can afford to have one. The alternative presented here is the doping technique, which has the important advantage that no towers are needed, thus enabling more researchers to manufacture their own doped fibers.
We have included the following sentences in the introduction in order to clarify the motivation of the work:
However, all these approaches are performed at the fiber preform level, which means that a fiber-drawing tower is needed to obtain the doped POFs. These towers are very expensive, so not all researchers can afford to have one. In contrast, this work describes a doping procedure…
With respect to the reviewer’s concern about the effect of having utilized a solvent to introduce the dopant molecules, we agree that the mechanical and optical characteristics of the optical fiber might be affected. However, Figure 6 shows that the attenuation coefficient does not change significantly because of the complete penetration of the solvent at 10 ºC. Furthermore, the influence of employing one type of dopant or another is probably much more important than the influence of having used a solvent, because the solvent is subsequently removed, whereas the dopant remains in the fiber.
3. If possible, please indicate the front thickness in Fig. 2
Following the reviewer’s recommendation, we have also included the values of the front thickness in the figure.
4. For the results of Fig. 3, it is better to use the similar scale for good comparison. In addition, please add the error bar for all the fitting results (similar for Fig. 4).
When we tried to plot all the figures with the same time scale, as the reviewer says, we realized that the evolution of both the dopant penetrations and the front thickness could not be observed clearly in the figures corresponding to higher temperatures. The main reason is that the necessary penetration times change greatly from one temperature to another. This is why we decided to plot each figure with different time scales.
On the other hand, following the reviewer’s recommendation, we have introduced the corresponding coefficient of determination for all the fittings in Table I.
5. What is the length of the sample used for this experiment? BTW, where you cited Figure 5?
The information about the absorption measurements is explained in the last paragraph of the section titled “Experimental setup”, where we explain that the employed fiber length was 1 cm for all the samples.
With regard to the citation of Figure 5 (now Figure 6), this is cited in the first paragraph of the section titled “Optical characterization”, after the reference [27].
6. It is not good to compare the efficiency by the wavelength. In fact, lifetime could be more related with the efficiency.
As far as we understand, the reviewer refers to the attenuation, because the only graphs plotted as functions of the wavelength are those of the attenuation. These are the most typical curves. Should the reviewer desire further explanations, we would appreciate a clarification of the question, because the word “efficiency” does not appear in our paper.
7. in Line 293-295, the author claimed that the reduction is smaller than other POF samples. If possible, please listed as Table for better comparison.
Following the reviewer’s recommendation, we have added the new Table 2.
8. The English is casual at some places (such as ‘Poly(methyl methacrylate)’, ‘reference signal power was’, ‘it might seem that the’, ‘The so obtained’, etc). Some font size is not right. The author should revise it carefully and thoroughly.
With the exception of Poly(methyl methacrylate), which is widely employed in the literature, we have modified the other expressions as follows:
“reference signal power” is now expressed as “reference signal”
“…it might seem that the use of high temperatures would be…” is now expressed as “…the use of high temperatures seems to be…”
“The so obtained attenuation values are very similar…” is now expressed as “The attenuation values obtained in the way described above are very similar…”
“…interest in the field of photonics has increased thanks to the incorporation…” is now expressed as “…interest in the field of photonics has increased as a result of the incorporation…”
Moreover, we have corrected all symbols with incorrect font size.
9. Some reference format/style is inconsistent (like 74, 9, 10, 23, 28). This should be revised and unified.
Following the reviewer’s recommendation, we have revised and corrected all the references according to the template provided by the journal.

Reviewer 2 Report
I suggest to accept the paper subject to a revision that would address the following issues:
The last sentence in the abstract is speculative and should be either removed or cofirmed.
Fig.5: at what wavelength was the attenuation measured?
Fig.7: please provide more detail on how the N1 and N2 z dependence measurement was performed. Was the fiber cut and light collected from the end or not and light collected from the side?
Fig.8: How was the light propagation distance de/increased? Again, by cutting the fiber?
Fig.9: How the normalised intensity was measured? Which wavelength range? How the light was collected from the fiber?
Also: Fig.1 provides the measurement setup schematic. I have difficulty with connecting this schematic with the results presented in Fig.5-Fig.9. Fig.1 seems not sufficently detailed.
Author Response
Manuscript ID: polymers-405534
Title: Fabrication of active polymer optical fibers by solution doping and their characterization
The changes in the paper are indicated below in red. The comments to the reviewers are written just above the changes and after each recommendation.
I suggest to accept the paper subject to a revision that would address the following issues:
Changes following the recommendations by the reviewer:
1. The last sentence in the abstract is speculative and should be either removed or confirmed.
According to the results obtained in this paper, and more specifically in the last paragraph of the text and in Figure 10, we can assure that the sentence holds true. Following the reviewer’s recommendation, we have rewritten the sentence of the abstract in the following way:
The samples prepared in the way described in this work present higher photostability than typical samples reported in the literature, which are prepared in different ways.
2. Fig.5: at what wavelength was the attenuation measured?
As has been written in the section titled “Experimental setup”, the absorption spectra of the fibers were measured by means of a Cary 50 UV-Vis spectrophotometer. This was adjusted to measure the absorption spectra in a continuous range of wavelengths from 270 nm to 900 nm. The emitted and received wavelengths are the same for each point of the range measured.
3. Fig.7: please provide more detail on how the N1 and N2 z dependence measurement was performed. Was the fiber cut and light collected from the end or not and light collected from the side?
Fig.8: How was the light propagation distance de/increased? Again, by cutting the fiber?
As is explained in the section titled “Experimental setup”, the evolutions of N1 and the square root of N2 as functions of the propagation distance were measured by using the side-illumination technique, in which each sample was excited laterally and the emitted light was collected from one of its ends by means of a spectrometer. In order to clarify this, we have modified the first sentence of the section in the following way:
Figure 1 sketches the experimental setup employed to measure the optical spectra by means of the side-illumination technique [20,21]. The POF sample is excited laterally and the light emitted from the excitation point propagates along the sample over a distance that can be adjusted by moving the linear stage employed to hold the sample. The light is finally collected by an optical spectrometer placed at the end of the sample. This is the procedure followed for all the measurements carried out as functions of either the propagation distance or the excitation time. The excitation light source was a…
4. Fig.9: How the normalised intensity was measured? Which wavelength range? How the light was collected from the fiber?
The light emitted by each sample was measured in the way just described in our previous answer.
In the case of the fluorescence intensities of our new Figure 10 (formerly called Figure 9), all the obtained spectra were integrated in wavelength. The results plotted in the figure were normalized to the value of 100% at the initial time of measuring process.
5. Also: Fig.1 provides the measurement setup schematic. I have difficulty with connecting this schematic with the results presented in Fig.5-Fig.9. Fig.1 seems not sufficently detailed.
We think that some of the comments of the reviewer can stem from our failure to explain Figure 1 in sufficient detail. The improved explanation has already been written in our answer to the comment labeled as “3” in this document.

Reviewer 3 Report
The manuscript by Ayesta et al. describes a method for the realization of active polymer fibers. The authors use so-called solution doping to incorporate the dopants directly into the core of the fibers. With this technique which seems to work remarkably well the authors prepare the fibers and characterize the doping efficiency for different temperatures. They find that the fibers exhibit better optical properties when lower temperatures are employed during the process. The authors draw the conclusion that the fibers prepared this way might show better photostability compared to fibers realized by different techniques which are reported in the literature. Also they hypothesize that these fibers might be relevant for all-optical devices.
The manuscript reports on interesting and certainly valid work on doping of polymer optical fibers. The paper is well written and understandable and the scientific case is presented in a clear way. Also the graphs are concise and reflect the aspects discussed in the text. That said I think that the manuscript deserves publication in Polymers and I can therefore recommend it for publication.
However the authors should address a few points prior to publication:
1. I think that the authors need to present more details on the possible applications of these fibers in the introduction and the conclusions section. Just stating that the fibers might be relevant for future applications is actually not enough and somewhat weak.
2. On page 9 the authors refer to the photostability of their fibers as compared to different samples, however, they do not give any numbers. This can easily be done and should be added.
3. Figure 6: presenting relative measurements is ok, but it would be more clear and also more open to present absolute numbers. I think it is ok if these numbers are very low but the readers deserve to know these details, in particular in the context of judging whether possible applications are realistic or not. I do not think that the paper becomes weaker by that, but stronger.
4. On page 6: the small holes which are formed are just described. It would be appropriate to show some microscopy pictures of these holes.
Author Response
Manuscript ID: polymers-405534
Title: Fabrication of active polymer optical fibers by solution doping and their characterization
The changes in the paper are indicated below in red. The comments to the reviewers are written just above the changes and after each recommendation.
The manuscript by Ayesta et al. describes a method for the realization of active polymer fibers. The authors use so-called solution doping to incorporate the dopants directly into the core of the fibers. With this technique which seems to work remarkably well the authors prepare the fibers and characterize the doping efficiency for different temperatures. They find that the fibers exhibit better optical properties when lower temperatures are employed during the process. The authors draw the conclusion that the fibers prepared this way might show better photostability compared to fibers realized by different techniques which are reported in the literature. Also they hypothesize that these fibers might be relevant for all-optical devices.
The manuscript reports on interesting and certainly valid work on doping of polymer optical fibers. The paper is well written and understandable and the scientific case is presented in a clear way. Also the graphs are concise and reflect the aspects discussed in the text. That said I think that the manuscript deserves publication in Polymers and I can therefore recommend it for publication.
However the authors should address a few points prior to publication:
Changes following the recommendations by the reviewer:
1. I think that the authors need to present more details on the possible applications of these fibers in the introduction and the conclusions section. Just stating that the fibers might be relevant for future applications is actually not enough and somewhat weak.
Considering the reviewer’s recommendation, we have added the following sentences in the introduction:
As the manufacturing temperature of POFs is much lower than that of glass fibers, it is possible to embed a wide range of dopants in the fiber core, from organic dyes and conjugated polymers to other kinds of materials, such as rare-earth ions and quantum dots. Some of these dopants can generate or amplify visible light at the low-attenuation windows of POFs [5,6]. The emission and absorption features of these dopants can be suitable for achieving luminescence in the visible region of the spectrum, which is interesting for a wide range of applications [7-9]. Consequently, research on doped POFs is on an upward trend. The results obtained are being applied for the development of optical sensors [7,8], of solar concentrators that collect and transport solar light [9], and of superluminiscent speckle-free light sources [5,10]. Additionally, the use of optical fibers allows for symmetrical output beams…
As can be seen in these sentences, we have introduced two new references:
[6] Parola, I.; Illarramendi, M.A.; Arrue, J.; Ayesta, I.; Jiménez, F.; Zubia, J.; Tagaya, A.; Koike, Y. Characterization of the optical gain in doped polymer optical fibres. J. Lumin. 2016, 8, 1-8.
[10] He, J.; Chan, W.-K.E.; Cheng, X.; Tse, M.-L.V.; Lu, C.; Wai, P.-K.A.; Savovic, S.; Tam, H.-Y. Experimental and theoretical investigation of the polymer optical fiber random laser with resonant feedback. Adv. Opt. Mater. 2018, 6, 1701187.
Furthermore, we have also added the following sentence in the conclusions:
This improvement, together with the rest of optical features and ease of fabrication, could be very interesting in the design process of all-optical devices based on active POFs, such as switches, lasers and amplifiers, optical sensors and solar concentrators.
2. On page 9 the authors refer to the photostability of their fibers as compared to different samples, however, they do not give any numbers. This can easily be done and should be added.
In the last paragraph of the paper, we mention that the photostability of our fibers was higher than those reported in references 36 and 19 for POF samples doped by means of the polymerization technique. Specifically, the photostability of our doped fibers was, respectively, 7 % and 18 % higher than those measured in the same conditions for some thermoplastic and thermosetting POFs. However, in order to present the improvement of the photostability more clearly, we have added the new Table 2, which summarizes the amount of degradation in each fiber while including more information than the previous version of the paper.
3. Figure 6: presenting relative measurements is ok, but it would be more clear and also more open to present absolute numbers. I think it is ok if these numbers are very low but the readers deserve to know these details, in particular in the context of judging whether possible applications are realistic or not. I do not think that the paper becomes weaker by that, but stronger.
Following the reviewer’s recommendation, we have modified Figure 7 (formerly called Figure 6) by plotting the absolute absorption coefficient. Accordingly, we have modified the figure caption as follows:
Absorption coefficient and emission spectrum corresponding to our POFs doped with rhodamine B by using the solution‑doping technique at 10 °C.
We have also modified the following sentence in the paragraph located just above the figure:
The absorption coefficient and the emission spectrum of these samples in the near-ultraviolet and visible regions are shown in Figure 7.
4. On page 6: the small holes which are formed are just described. It would be appropriate to show some microscopy pictures of these holes.
Following the reviewer’s recommendation, we have included a new figure (Figure 5) to compare the external appearance of two fiber samples doped at different temperatures: at 10 ºC and at 35 ºC. As can be seen, the formation of holes is very clear at 35 ºC.
Furthermore, we have modified the following sentences in the first paragraph of the section titled “Optical characterization”:
The analysis of the doping diffusion at different temperatures revealed that the translucency of the doped material was much lower when the doping process was carried out above room temperature (i.e. at any of the temperatures equal to or greater than 35 °C in Figure 3), foreshadowing bad light-transmission features. This effect is clearly noticeable in Figure 5, and it is a direct consequence of the interaction between the methanol solvent and the PMMA of the fiber core…

Reviewer 4 Report
The paper presents interesting results on the development of active polymer fibers by solution doping by rhodamine B.
Despite the fact that there are similar works towrds post processing of optical fibers by solution doping the paper contains a useful analysis and extensive characterization that could be of interest to journal's readers.
Authors should better and in more detail discuss the difference of current work with Ref 13, as the solution doping is applied in both cases in fibers and not in performs.
Although in [13] is claimed the doping was performed only in cladding there is a study in the doping of the entire fiber down to the core. Please state the differences with current work.
Furthermore in the current study the solution doping procedure is applied in uncladded fiber. How the lack of cladding region affects the doping dynamics?
How this un-cladded structure affects the propagation characteristics if compared to a cladded fiber ?
Since the present study is essentially targeted to low cost ad-hoc doping procedure authors should comment on this simplified core only structure and its limitation in real applications.
Please provide also the refractive index profile of the doped fiber, or comment on this.
Authors should give some characterization results on light guiding of the developed fibers
Furthermore Authors should include the first two sections in a Materials and Methods main section.
I would suggest also to replace the word "virgin fiber" with "pristine fiber as it more commonly used in such cases.
Author Response
Manuscript ID: polymers-405534
Title: Fabrication of active polymer optical fibers by solution doping and their characterization
The changes in the paper are indicated below in red. The comments to the reviewers are written just above the changes and after each recommendation.
The paper presents interesting results on the development of active polymer fibers by solution doping by rhodamine B.
Despite the fact that there are similar works towards post processing of optical fibers by solution doping the paper contains a useful analysis and extensive characterization that could be of interest to journal's readers.
Changes following the recommendations by the reviewer:
1. Authors should better and in more detail discuss the difference of current work with Ref 13, as the solution doping is applied in both cases in fibers and not in performs.
Although in [13] is claimed the doping was performed only in cladding there is a study in the doping of the entire fiber down to the core. Please state the differences with current work.
Furthermore in the current study the solution doping procedure is applied in uncladded fiber. How the lack of cladding region affects the doping dynamics?
Certainly, the solution doping was also applied for POFs in reference [15] (old reference [13]), in which the authors also studied the penetration of the solvent down to the center of their cores. However, they employed commercial cladded POFs and, therefore, the interface between the core and the cladding acted as a barrier that prevented the penetration of the dopant molecules into the core, while methanol could penetrate. In contrast, the samples employed in our work, in the absence of cladding, allowed the complete penetration of both the dopant molecules (rhodamine B) and the solvent (methanol), thus enabling us to carry out a thorough characterization of the doping process. Furthermore, the dopant molecules could be made to penetrate to the center of the core. Besides, POFs doped in the core are usually preferred for active POFs, rather than POFs only doped in the cladding.
We have incorporated these ideas to the introduction in the following way:
Until now, few works employing the solution-doping technique in POFs have been reported, and only the preform or the fiber cladding were doped [14,15]. In [15], cladded POFs were employed. The dopant molecules could not thereby penetrate into the core, because the interface between the core and the cladding acted as a barrier for them, while the solvent (methanol) could penetrate. Therefore, the cladding was the only doped region. However, following the procedure described in this paper, we can obtain POFs whose core is completely and uniformly doped with active molecules.
2. How this un-cladded structure affects the propagation characteristics if compared to a cladded fiber? Since the present study is essentially targeted to low cost ad-hoc doping procedure authors should comment on this simplified core only structure and its limitation in real applications.
We agree with the reviewer that only-core fibers have more limitations than cladded ones in real applications, because the propagation features are negatively affected by the absence of cladding, and also because the core material is more prone to being affected by external unwanted agents. Although the doping procedure was always applied to only-core POFs in this work, in a second stage we incorporated a cladding layer of a polymeric material to the best fibers, namely those that were doped at 10 ºC, this temperature having resulted to be the best one in terms of low attenuation. It should be noted that all of the optical features presented in this work were measured from doped POFs with cladding. In order to emphasize this idea, we have modified the following sentences in the section titled “Optical characterization”:
Taking the aforementioned results into account, samples of 12 cm were fabricated at the suitable temperature of 10 °C. Moreover, in a second stage we incorporated a cladding layer of around 10 µm of a polymeric material to these optimum samples, in order to improve the light transmission features. Besides, the core material is less prone to being affected by external unwanted agents if a cladding is added. This process was carried out in the way described in the section about the preparation of the samples.
3. Please provide also the refractive index profile of the doped fiber, or comment on this.
Following the reviewer’s recommendation, we have added the following sentence in the first paragraph of the section titled “Sample preparation”:
… which enabled us to control the fiber diameter with an accuracy of ±1 % [16]. The refractive-index profile of the fibers was step-index in all cases. Alternatively, there are only-core POFs commercially available …
4. Authors should give some characterization results on light guiding of the developed fibers.
In this paper, we carried out an optical characterization of our doped fibers when they were excited at points located at different distances from the fiber end, thereby allowing us to observe the evolution of the emitted light as a function of the propagation distance. From this information, we were able to obtain the linear attenuation coefficients at different wavelengths (from 570 nm to 650 nm). This parameter indicates the rate at which light is attenuated as it propagates along the fiber. Moreover, we have compared the obtained attenuation coefficients with those reported for other types of fibers, such as thermosetting ones and thermoplastic ones fabricated differently, by means of the gel-polymerization technique. We consider that a more comprehensive characterization would lengthen this paper excessively. Furthermore, we are considering writing another paper dedicated to the characterization of our doped fibers when the excitation energy is above the threshold value.
5. Furthermore Authors should include the first two sections in a Materials and Methods main section.
Following the reviewer’s recommendation, we have created a section titled “Materials and methods”, in which we have introduced the subsections titled “Sample preparation” and “Experimental setup”.
6. I would suggest also to replace the word "virgin fiber" with "pristine fiber” as it more commonly used in such cases.
Following the reviewer’s recommendation, we have replaced the word “virgin fiber” with “pristine fiber”.

Round 2
Reviewer 1 Report
All my concerns have well been responded and recommended the acceptance of the publication by Polymers.
Reviewer 4 Report
Authors have considered the comments adequately and in the revised version they have clarified several issues. I suggest the acceptance of the paper.